





# Satellite Data Rendered Irrigation using Penman-Monteith and SEBAL - sDRIPS for Surface Water Irrigation Optimization

Shahzaib Khan<sup>1</sup>, Faisal Hossain<sup>1</sup>, Khairul Islam<sup>2</sup>, Mahfuz Ahamed<sup>2</sup>

5 Department of Civil and Environmental Engineering, University of Washington, Seattle, WA 98915, USA

<sup>2</sup>Bangladesh Water Development Board, Pani Bhavan, Dhaka, Bangladesh

Correspondence to: Shahzaib Khan (skhan7@uw.edu)

Abstract. This study proposes a satellite remote sensing-based water-provider-centric irrigation advisory system designed to manage surface water resources and allocate water efficiently to areas in need, thereby promoting sustainable irrigation practices in the context of a changing climate. The system utilizes satellite remote sensing based SEBAL (Surface Energy Balance Algorithm for Land) and Penman-Monteith evapotranspiration models to estimate crop water use. By integrating the responses from the previous irrigation cycle, current precipitation, forecasted precipitation, and evapotranspiration-based water needs, the framework calculates the net water requirements for command areas within irrigation canal networks. Operating on a weekly basis, the system generates advisories that enable the irrigation water provider to make informed, science-based decisions about water allocation. These advisories quantify the net water requirement, giving water providers the flexibility to dispatch water to areas of higher need based on their on-ground judgment. Additionally, the proposed framework can simulate future cropping patterns by assuming potential policy changes or net reduction in water supply in the main canal due to climate change or increased transboundary withdrawal. The advisory system is co-developed and implemented with the irrigation management agency called Bangladesh Water Development Board on the Teesta River Irrigation System located in Northern Bangladesh. The study demonstrates its effectiveness when compared against actual water supplied for irrigation. However, the application of sDRIPS is not limited to Bangladesh, as it is scalable to other regions with similar water management challenges for agriculture.

**Keywords:** Irrigation, Surface Water Management, Optimization, Satellite Remote Sensing, Cloud Computing, Teesta River, Bangladesh

## 1. Introduction

Freshwater is an essential resource for human survival and underpins a wide range of economic activities. Among the various sectors that are dependent on freshwater, agriculture is the largest consumer, accounting for approximately 85% of all human freshwater water consumption (D'Odorico et al., 2020). Projections for the year 2050 indicate a global population increase to more than 9.0 billion, which will further exacerbate the pressure on existing freshwater resources for food production, particularly groundwater (Boretti & Rosa, 2019). Recent studies have highlighted the significant depletion of groundwater resources in major food-producing regions and other areas worldwide (Feng et al., 2013; Jasechko et al., 2024; Kuang et al., 2024; Liu et al., 2022). In response to this issue, there is an urgent need for the conjunctive and optimal use of groundwater and surface water resources. Nations such as India, Pakistan, and Bangladesh

https://doi.org/10.5194/egusphere-2025-4574 Preprint. Discussion started: 16 October 2025 © Author(s) 2025. CC BY 4.0 License.






have increasingly promoted the use of surface water resources to alleviate pressure on groundwater reserves (Paul & Hasan, 2021). To enhance the effectiveness of these efforts, systematic and frequent water accounting is essential. This process should comprehensively account for direct human water use—including agricultural, municipal, commercial, and industrial consumption, as well as indirect losses due to reservoir evaporation and evapotranspiration (Richter et al., 2024). Such analyses are critical for quantifying stress on different water sources and informing strategies for their sustainable and efficient management to meet growing water demands.

Assuming a scenario where both surface water and groundwater resources are sufficient to meet food production needs, and we have a robust infrastructure for supplying from both sources, surface water is the more sustainable source due to its multifaceted benefits. Firstly, surface water often contains higher nutrient levels than groundwater, which can lead to increased crop yields. Secondly, by reducing the reliance on groundwater pumping, surface water allows aquifers to recover naturally by acting as a "storage battery" during dry seasons and ensuring a long-term water supply for agricultural and other uses. Thirdly, and perhaps most importantly, surface water irrigation systems are mostly gravity-driven and require less energy than groundwater pumping systems. This reduction in energy consumption translates to lower greenhouse gas emissions, thereby reducing the carbon footprint of agricultural practices (Qin et al., 2024). Accounting for government policies, over-exploitation of existing irrigation water resources, and the benefits of surface water irrigation, there is a need to manage surface water resources sustainably and efficiently.

To manage surface water irrigation more efficiently, adapting traditional irrigation practices to address the challenges posed by climate variability and increasing demand is crucial. Rising temperatures and changing rainfall patterns significantly impact crop growth and surface water availability. This in turn, demands that current surface water irrigation practices become more optimized. Studies led by Kovenock and Farhat (2018, 2021) highlight the complex relationship between rising temperatures and biomass production. Kovenock & Swann (2018) have reported that increased carbon dioxide (CO<sub>2</sub>) concentrations can negatively affect biomass productivity and alter the leaf area, which reduces the resilience against climate change. Farhat et al. (2021) showed that the elevated temperature leads to a higher uptake of toxic substances in groundwater such as arsenic. This bioaccumulation eventually results in plant failure at elevated temperatures, affecting both food quality and quantity. In many places, such as in Bangladesh where groundwater can often be contaminated by arsenic (Pogorski and Berg, 2020), the surface water usually has a better quality for irrigation applications. Overall, surface water can present a better alternative, in many places, in the light of climate change, for mitigating these adverse effects on food production in terms of quality and sustainability.

To fully leverage the benefits of surface water and adapt to climate variability, it is essential to overcome the limitations of traditional irrigation practices. Traditional-irrigation-practices specifically refer to those employed by the water-providers, encompassing temporal, intensity-based, and spatial components of irrigation decision-making. These components are broken down into the following:

 When to irrigate (time - T) – determining whether water should be distributed in the current cycle based on the combined effects of the previous irrigation cycle, precipitation response, and crop water need;

- How much water is needed (intensity L³) evaluating the volume of water required by analyzing
  the intensity of the combined effects of the previous irrigation cycle, precipitation events, and crop
  water need; and
  - 3. Which region to irrigate (space Location) identifying regions requiring irrigation by assessing the spatial distribution of the previous irrigation cycle, precipitation events, and crop water need.
- 90 In reality, traditional irrigation practices are not limited to water-providers but extend to water-consumers (farmers), who rely heavily on traditional irrigation calendars. These calendars, which are static and developed based on the historical climatology of the previous century assume a fixed cropping pattern. These calendars inform farmers of the irrigation schedule to follow but are becoming increasingly inadequate in the face of rising food production demands. The reliance on these static crop calendars leads 95 to inefficiencies in water management and reduced agricultural productivity, as evidenced in many countries such as India, Bangladesh, Pakistan, and Australia (Bose et al., 2021; Bretreger et al., 2020; Hossain et al., 2017). Regardless of the region, farmers typically are not adaptive to rapid climate variability and cannot accurately predict precipitation events and their amounts at sub-seasonal irrigation timescales. This uncertainty often leads to overwatering of the fields to avoid the risk of crop loss due to rising 100 temperatures. Overwatering, however, can inadvertently wash away essential soil nutrients, while underwatering can stress crops, both resulting in reduced yields. These practices lead to inefficiencies in water use and diminish soil fertility and agricultural productivity (Hossain et al., 2017).
- Considering the effects on agricultural yield due to altered rainfall patterns and rising temperatures, there is an urgent need for a scientific-data-driven approach to optimize existing irrigation practices to achieve maximum crop water productivity. The potential of using satellite data for sustainable agriculture and efficient irrigation has been widely acknowledged in recent years (Deines et al., 2019; Sishodia et al., 2020; Zhang et al., 2022). Several studies have employed crop coefficients (Kc) based evapotranspiration techniques to enhance the understanding of irrigation schedules and irrigate fields based on actual water needs (Bretreger et al., 2020; Gabr & Fattouh, 2021). Tools like the Food and Agriculture Organization's (FAO) CROPWAT, which utilizes the Penman-Monteith equation (Allen et al., 1998), have provided valuable insights into regional water requirements (Dong, 2018; Khaydar et al., 2021; Solangi et al., 2022; Surendran et al., 2015).
- 115 Despite significant advancements, substantial gaps remain in the current body of research that need to be addressed. These gaps arise from limitations in the temporal and spatial resolution of past studies. The trade-off between temporal and spatial resolution of satellite data makes it challenging to estimate evapotranspiration (ET) at high temporal and spatial resolutions simultaneously. Typically, temporal resolution ranges from daily to monthly time steps. When the temporal resolution is one day, the spatial 120 resolution is usually at the kilometer scale (Chen et al., 2021). Conversely, if a medium spatial resolution sensor having 30-meter pixels is selected, the temporal resolution is typically around 16 days from polarorbiting optical satellites (Tasumi, 2019). To achieve an optimal balance between spatial and temporal resolution, a combination of satellites needs to be used. Additionally, challenges exist in operationalizing findings or tools due to the latency of the available latest meteorological data. For instance, the Global Land 125 Data Assimilation System (GLDAS), which provides global meteorological forcing data, can be used in estimating evapotranspiration (Pareeth & Karimi, 2023), has an early product stream with a latency of about 1.5 months. This latency hinders the ability to use these datasets for near real-time operational studies.







Tools available for surface water irrigation also suffer from an inability to accurately distinguish between the actual water provided to plants and the water needed by plants. This arises from the difficulty in estimating ET under ambient water stress conditions for each crop and lack of metering and monitoring. The incomplete information makes it difficult to estimate net water requirements and project potential water savings. Additionally, there is a lack of integration of precipitation forecasts at the operational level to optimize irrigation even further. It is important to acknowledge that, to the best of our knowledge, there is no freely available open-source tool that addresses these gaps in optimizing surface water irrigation schemes based on satellite data.

In this study, we present Satellite Data Rendered Irrigation using Penman-Monteith and SEBAL (sDRIPS., pronounced as "drips"), a cloud-based optimized surface water irrigation advisory system that leverages earth observations and weather models. sDRIPS. utilizes publicly available data from the Landsat satellite series, the Global Forecast System (GFS), and the Global Precipitation Measurement's (GPM) IMERG data (precipitation dataset). The founding concept of sDRIPS originated from the Irrigation Advisory System (IAS) first prototyped by Hossain et al. (2017). Later, Bose et al. (2021) integrated Gravity Recovery and Climate Experiment (GRACE) satellite-based water storage data with Landsat data to quantify the impact of IAS. Bose et al. (2021) also used satellite and model weather data to estimate ET using the Penman-Monteith method (Allen et al., 1998) and the Surface Energy Balance Algorithm for Land (SEBAL; Bastiaanssen et al., 1998) as proxies for crop water demand and actual water consumed, respectively.

The core concept of IAS has been implemented in various regions under different names, such as IAS for Pakistan (Hossain et al., 2017), Provision for Advisory on Necessary Irrigation (PANI) in India (Hossain et al., 2020), and Integrated Rice Advisory System (IRAS) in Bangladesh (Hossain et al., 2022). IRAS represented a significant technological leap in achieving higher spatial resolution from the km scale of IAS to the 30m scale using Landsat Thermal Infrared (TIR) data to estimate actual ET based on SEBAL. Combined with crop water demand from Penman-Monteith FAO 56 (Allen et al., 1998) that could already be estimated at comparable scales, IRAS offered the prediction of potential over or under-irrigation based on the comparison of actual ET estimates from SEBAL and crop water demand estimates from Penman-Monteith (Bose et al., 2021). Additional improvements in IRAS include cloud-based operationalization and automation that requires minimum internet bandwidth or local computing resources for the stakeholder agency. Climatic variables like precipitation, temperature, and wind speed are incorporated to generate advisories for farmers on how much they should water their crops based on potential over or under-irrigation in the previous week (Hossain et al., 2022). These advisories are then automatically transmitted to relevant users comprising farmers, irrigation district managers and water agency staff. With the combined use of two Landsat missions 8 and 9, the temporal resolution of IRAS has been improved from biweekly to weekly frequency. Additionally, dynamic crop coefficient values based on planting dates are used in IRAS to accurately estimate the actual water need for a given date.

Due to these innovative features, IRAS has been adopted by the Department of Agricultural Extension (DAE), of the Ministry of Agriculture of Bangladesh (<a href="https://iras.bamis.gov.bd/">https://iras.bamis.gov.bd/</a> or visit <a href="https://bamis.gov.bd">https://iras.bamis.gov.bd/</a> or visit <a href="https://bamis.gov.bd">https://bamis.gov.bd</a> and click "Satellite Products"). Currently, IRAS operates on DAE servers, providing irrigation advisories to farmers across approximately 2,000 agricultural districts located in the Northwestern and Northeastern regions of Bangladesh (Landsat Science, 2023).








Given the recent adoption of the farmer/user-centric advisory system of IRAS that optimizes water use at the consumer side, we believe there is now an opportunity for an operator-centric or water-provider-centric irrigation advisory system for surface water irrigation at the supply side. This proposed operator-centric (hereafter alternated with 'provider-centric') advisory can potentially guide irrigation water managers to optimize water delivery at the supply side where water is being centrally withdrawn from a river for dispatch to the irrigation command areas. Water suppliers and canal operators can be informed as to how much water should be allocated or delivered to each region or canal based on the water needs of all the farmers of that region. By assimilating nowcast and forecast precipitation data that are now available from weather forecast systems with the previous week's water allotment to the region and the current crop water demand, an accurate estimate of net water requirement for each region can be estimated. This net water requirement can be translated to the flux of surface water (L³/T) that would need to be maintained in water-supplying canals at the supply side for water managers. Consequently, such a system can support dispatch decisions for surface water irrigation based on data-driven science, crop water needs, and ambient environmental conditions.

In this study, our central research question is: *How can we dynamically optimize a multi-branched canal irrigation system using earth observation satellites?* We believe that addressing the sustainable use of surface water requires tackling it from both the consumer side (by farmers/users) and supply side (by water managers/canal operators) with an operator-centric system as a complement to the user-centric system. The key goal of this study is to guide canal operators and help them make informed decisions on how much surface water to dispatch centrally from the source at the river for the command areas to maximize agricultural water productivity. We developed and implemented sDRIPS for the Teesta River Irrigation system located in Northern Bangladesh where there is currently an urgent need to optimize the use of surface water resources.

# 2. Study Area

The sDRIPS study focuses on the Teesta Barrage Project (TBP) in Bangladesh, the country's largest surface water irrigation project. Established in 1990, the TBP supplies irrigation water from January to April and spans the Teesta River at Dalia-Doani Point in the Lalmonirhat district (Fig. 1 Middle Panel). This project features a 615-meter-long concrete structure equipped with 44 radial gates, providing a discharge capacity of 12,750 cubic feet per second and supporting a command area of 154,250 hectares through a 4,500-kilometer network of canals (River Research Institute, 2023). For more details on TBP, readers are referred to (River Research Institute, 2023).

The main TBP canal divides into three primary canals: Dinajpur Canal, Rangpur Canal, and Bogra Canal (Fig. 1 Middle Panel), which further branches into an intricate network of secondary and tertiary canals. Selecting the TBP as a study site offers two primary advantages: the utilization of in-situ water supply data at the head canal and the opportunity to study a complex canal network system. A successful implementation of our study to implement sDRIPS for TBP here means it can be adapted and applied to surface water irrigation systems around the world. For this study, irrigation canal shapefiles, command areas, and in-situ water supply data were provided by the Bangladesh Water Development Board (BWDB).



Fig. 1: The Teesta Barrage Project (TBP). The left panel displays the map of Bangladesh with the location of TBP on the Teesta River. The middle panel illustrates the network of irrigation canals. The right panel highlights the command areas served by the primary, secondary, and tertiary canals. Satellite basemap from Bing Maps (© Microsoft).

## 3. Datasets Used

The publicly available datasets used in sDRIPS allow for accurate assessments of the actual water needs of command areas while considering climatic factors. To facilitate implementation and reduce technical barriers, the cloud computing platform Google Earth Engine (GEE), developed by Gorelick et al. (2017), was utilized. The cloud computing platform helps sDRIPS to eliminate the need to download and process satellite and weather data on the local machine. Table 1 provides a comprehensive list of the datasets utilized by sDRIPS. Some of the key datasets, which are referenced by their initials in Table 1 but not previously mentioned in the text are - Shuttle Radar Topography Mission (SRTM), Global Land Cover Characterization (GLCC), and National Oceanic and Atmospheric Administration (NOAA).

Table 1: Datasets used in sDRIPS

| Dataset                               | Dataset ID on<br>GEE | Band<br>Name/Derived<br>Products             | Description              | Spatial<br>Resolution | Temporal<br>Resolution |
|---------------------------------------|----------------------|----------------------------------------------|--------------------------|-----------------------|------------------------|
| Global<br>Forecasting<br>System (GFS) | NOAA/GFS0P25         | temperature_2m_a<br>bove_ground              | Air Temperature          | 25 km                 | 6 hours                |
|                                       |                      | u_component_of_<br>wind_10m_above<br>_ground | Wind Speed (u component) |                       |                        |
|                                       |                      | u_component_of_<br>wind_10m_above<br>_ground | Wind Speed (v component) |                       |                        |

|                                                  |                                                                | specific_humidity<br>_2m_above_groun<br>d | Specific Humidity                                                       |       |                                |
|--------------------------------------------------|----------------------------------------------------------------|-------------------------------------------|-------------------------------------------------------------------------|-------|--------------------------------|
|                                                  |                                                                |                                           | Pressure<br>(Estimated using<br>Hypsometric<br>Equation)                |       |                                |
|                                                  | GEE and NOAA                                                   | total_precipitation<br>_surface           | 168-hour<br>Precipitation                                               |       |                                |
| Landsat 8 and<br>Landsat 9 satellite<br>series   | LANDSAT/LC08/<br>C02/T1_TOA and<br>LANDSAT/LC09/<br>C02/T1_TOA | B2                                        | Blue                                                                    | 30 m  | 16 days                        |
|                                                  |                                                                | B4                                        | Red                                                                     |       | individual, 8 days<br>combined |
|                                                  |                                                                | B5                                        | NIR                                                                     |       |                                |
|                                                  |                                                                | В6                                        | Shortwave IR                                                            |       |                                |
|                                                  |                                                                | B10                                       | Low Gain<br>Thermal                                                     |       |                                |
|                                                  |                                                                | B11                                       | High Gain<br>Thermal                                                    |       |                                |
| Sentinel 1 satellite                             | COPERNICUS/S<br>1_GRD                                          | VV                                        | Single co-<br>polarization,<br>vertical<br>transmit/vertical<br>receive | 10 m  | 10 days                        |
| Shuttle Radar<br>Terrain Mapping<br>(SRTM)       | USGS/SRTMGL1<br>_003                                           |                                           | Digital Elevation<br>Map                                                | 30 m  |                                |
| Global Land<br>Cover<br>Classification<br>(GLCC) | COPERNICUS/L<br>andcover/100m/Pr<br>oba-V-C3/Global            | discrete_classifica<br>tion               | Land Use Land<br>Cover                                                  | 100 m |                                |
| IMERG GPM                                        |                                                                |                                           | Precipitation                                                           | 10 km | Around 4 hours                 |

# 4. How sDRIPS Works


An overview of sDRIPS is first provided along with an illustration (Fig. 2), followed by a brief description of the steps of the processes. These steps are then covered in detail in the methodology section.

Fig. 2: Illustration of how sDRIPS works. Green represents the regions where surplus irrigation has been provided, while red regions are deficit regions needing more irrigation.

When a Landsat satellite overpasses a command area, sDRIPS collects crucial information required to estimate the actual water needs of the crop based on its growth stage using Penman-Monteith ET, and the amount of water applied to the crop using SEBAL ET. The impact of precipitation is also considered to refine the water demand estimation. This includes both nowcast precipitation data and forecasts of incoming precipitation. The system then compares these estimates to determine whether the water needs of all crops in the given command area are met. If the net water need is not satisfied, the area is classified as a deficit region, and the volume of water needed for dispatch for the coming week is quantified (Fig. 2). Conversely, if the water demand is met and there is an excess of provided water, the area is classified as a surplus region, The volume of surplus water that can be potentially stored locally and used later for nearby deficit regions is quantified (Fig. 2). This quantification allows water providers to monitor and adjust the water supply and dispatch decisions from the main canal efficiently to meet the specific needs of each command area.

Fig. 3 illustrates the proposed methodology for the irrigation advisory system, sDRIPS. Before starting, users need to have a GEE account (<a href="https://signup.earthengine.google.com/">https://signup.earthengine.google.com/</a>) to estimate Penman–Monteith and SEBAL evapotranspiration and a Precipitation Processing System (PPS) account (<a href="https://registration.pps.eosdis.nasa.gov/registration/">https://registration.pps.eosdis.nasa.gov/registration/</a>) to download the GPM IMERG precipitation data.

Fig. 3: Flowchart illustrating the proposed data processing methodology for the irrigation advisory system.

- Step 1: Create command areas based on the availability of command area data: This includes creating configuration files for the command areas and updating the script configuration files accordingly to run the system.
  - **Step 2: Estimate Penman-Montieth and SEBAL ET:** Use GEE to calculate evapotranspiration using the Penman-Monteith method and SEBAL.
- Step 3: Estimate net water requirements for each command area: Estimating soil moisture and percolation using Sentinel 1. Download IMERG precipitation data and use it to estimate the net water requirements by combining past precipitation with GFS precipitation forecasts.
  - **Step 4: Estimate the water supplied to each command area:** Calculate the water supply at the head canal and distribute it across the command areas based on the supply.
- Step 5: Assess irrigation status and generate plots: Evaluate the irrigation status using the estimated ET and water supply and generate water stress plots for analysis.

# 5. Methodology

# 5.1. Creation of Configuration Files (Step 1)

The configuration files provide the essential information required for the sDRIPS system to tailor the advisory system to the specific needs of the user. Readers are requested to refer to Section S1 of the supplementary material for more details.

285

295

300

# 5.2. Penman-Monteith and SEBAL Based Evapotranspiration Estimation (Step 2)

After the creation of the configuration files, the next step is to estimate Penman-Monteith and SEBAL-based evapotranspiration using Landsat data. sDRIPS checks the latest Landsat imagery from both Landsat 8 and Landsat 9 collections for each command area at the specified time. Once an image for the given area is found, the timestep of the image is noted and used to estimate the Kc value by calculating the difference between the planting date and the image timestep and using the crop configuration file. Similarly, other useful information, as mentioned in Table 1, is stored temporarily on the cloud.

### 5.2.1. Penman-Monteith Evapotranspiration

Penman-Monteith evapotranspiration (ETo) serves as a proxy for potential water demand for a reference crop. ETo is estimated for the Landsat overpass date and for all command areas following the methodology of Allen et al., (1998). The equation for ETo is as follows:

$$ETo = \frac{0.408\Delta(R_n - G) + \gamma \frac{900}{T + 273} u_2(e_s - e_a)}{\Delta + \gamma(1 + 0.34u_2)}$$

Where ETo is reference evapotranspiration (mm/day),  $R_n$  is net radiation at the crop surface (MJ/m²/day), G is soil heat flux density (MJ/m²/day), T is mean daily air temperature at 2 m height (°C),  $u_2$  is wind speed at 2 m height (m/s),  $e_s$  is saturation vapor pressure (kPa),  $e_a$  is actual vapor pressure (kPa),  $e_s$ — $e_a$  is saturation vapor pressure deficit (kPa),  $\Delta$  is slope of the vapor pressure curve (kPa/°C),  $\gamma$  is psychrometric constant (kPa/°C).

ETo calculated using

1 is the evapotranspiration for a reference crop. It is then adjusted for the actual crop growing in the given command area by considering the crop type, development stage, and water stress on crop transpiration for that command area using the following equation

$$ET = ETo \times K_c \times K_s$$

Where ET is the potential evapotranspiration (mm/day), K<sub>c</sub> is the crop coefficient and K<sub>s</sub> is the soil water stress coefficient. Unless otherwise specified, hereafter, PET will be used as a term to refer to crop water demand ET based on the Penman-Monteith equation.

#### 5.2.2. SEBAL Evapotranspiration

In our study, we employed the SEBAL algorithm developed by Bastiaanssen et al. (1998). The SEBAL model has been effectively implemented in more than 30 countries (Bastiaanssen et al., 2005). The SEBAL model has demonstrated high accuracy in estimating evapotranspiration. A few examples are - Zahid et al. (2023) compared SEBAL ET with lysimeter ET in Pakistan, finding the Coefficient of Determination (R<sup>2</sup>)





of 0.9 and a Root Mean Square Error (RMSE) of 1.26 mm/day with maximum error as 24.3%. Similarly, Zoratipour et al. (2023) found an  $R^2$  of 0.9 and an RMSE of 2.15 mm/day in Iran, and Rawat et al. (2017) validated SEBAL ET estimates in India with an  $R^2$  of 0.91. Typically the accuracies of the estimated SEBAL ET model are 85%, 95%, and 96% at daily, seasonal, and annual scales, respectively (Tang et al., 2013). SEBAL used in the sDRIPS framework was originally employed and validated by Bose et al. (2021) and Hossain et al. (2022).

The SEBAL model solves the surface energy balance equation to estimate evapotranspiration using satellite images and meteorological forcings data. SEBAL computes an instantaneous ET flux for the Landsat overpass time. A series of equations are incorporated in the SEBAL model that computes net surface radiation, soil heat flux, and sensible heat flux to the air. The residual energy flux is then calculated by subtracting the soil and sensible heat fluxes from the net radiation at the surface. This residual energy (latent heat) enables liquid water to transition to the water vapor phase, that is, the required evapotranspiration. Thus, for each pixel of the image, the ET flux is calculated as a residual of the surface energy budget shown in Equation 3.

$$\lambda E = Rn - G - H$$

Where  $\lambda E$  is latent heat flux (energy used for evapotranspiration),  $R_n$  is net radiation at the surface, G is soil heat flux, and H is sensible heat flux. SEBAL-based evapotranspiration (SEBAL ET) serves as a proxy for the actual water consumed by the crop, a concept validated by Bose et al. (2021). In our study, daily or 24-hour evapotranspiration was computed by assuming that variations in instantaneous evapotranspiration are not significant over the 24-hour period (Allen et al., 2007). For weekly evapotranspiration calculations, we estimated evapotranspiration on the day of Landsat acquisition and considered this steady-state evapotranspiration value for the next seven days until the availability of the next Landsat image. In this study the term SEBAL ET will refer to actual ET estimated using the SEBAL method (Equation 3)

#### 5.2.3. Deficit Irrigation

To quantify the evapotranspiration-based water requirements from a command area, we estimate the deficit for each command area or region. The deficit was calculated using the Equation 4.

$$Deficit Irrigation = SEBAL ET - PenmanMonteith ET$$

Fig. 4 Panel (a) represents the cumulative sum of Penman-Monteith based ET for 7 days, depicting the total water demand based on the rice crop water requirements. Similarly, Panel (b) is the cumulative sum of SEBAL-based ET for 7 days, providing an estimate of the actual water consumed by the crops, factoring in the actual conditions of the fields. Panel (c) shows the deficit and sufficient regions. The intensity of the deficit or sufficient is estimated using Equation 4 for every pixel. The deficit regions indicate areas where the water demand exceeds the supplied water, while sufficient regions indicate areas where the supplied water meets the demand. This detailed mapping allows water managers to identify critical areas needing







more precise water management interventions. Additionally, an uncertainty analysis of the SEBAL ET estimates was performed using Monte Carlo simulations, revealing a normalized deviation of 8.6% from the ensemble mean for the operational period of TBP. For further details, please refer to Section S3 of the supplementary materials.

Fig. 4: Map of estimation of ET-based deficit and sufficient irrigation regions.

## 5.3. Estimation of Net Water Requirement (Step 3)

To accurately determine the net water requirement, several factors must be considered, with soil moisture being a critical component. Soil moisture plays a pivotal role in understanding the interactions between surface and groundwater and in generating accurate irrigation advisories, as it governs the amount of water percolating through the soil. Water lost through percolation, which moves beyond the crop root zone, becomes unavailable for crops. To account for percolation in our framework, we utilized Sentinel-1 Synthetic Aperture Radar (SAR) data available on GEE. While global soil moisture products have been developed at coarser resolutions ranging from 9 km to 40 km (Chan et al., 2016; Kerr et al., 2012; Kim et al., 2023), these products are unsuitable for field-scale applications. SAR sensors, on the other hand, achieve finer spatial resolutions (10 m), making them suitable for agricultural field-scale analysis (Arias et al., 2023). Sentinel-1 C-band (wavelength of 5cm) data has been extensively used in soil moisture estimation studies and has demonstrated promising results up to 100 mm depth (Arias et al., 2023; Bauer-Marschallinger et al., 2019; Bhogapurapu et al., 2022; Wagner et al., 1999). While ground sensors provide ideal soil moisture measurements, installing them is often impractical on a large scale. To maintain the global scalability of the framework, Sentinel-1 C-band data was selected as the most viable alternative for estimating soil moisture. After estimating soil moisture from Sentinel-1 data, soil moisture at field capacity was derived using the Hengl & Gupta, (2019) dataset available on GEE. Percolation was subsequently calculated using Equation 5.

$$Percolation = Soil Moisture - Field Capacity$$
 5

Fig. 5 Panel (a) shows the average soil moisture estimated using the Sentinel 1 C band for each command area, Panel (b) shows the soil moisture at field capacity from Hengl & Gupta, (2019) and Panel (c) shows the average percolation that happened in each command area. Since percolation only happens when the soil moisture is greater than the soil moisture at field capacity, percolation was estimated using the following equation (Equation 6) for each pixel and then averaged over the command area.

$$Percolation = \begin{cases} Soil\ Moisture\ - Field\ Capacity\ ; \ge 0 \\ 0;\ else \end{cases}$$

Fig. 5: The left panel shows the average soil moisture estimated using Sentinel 1 C band on 6th March 2023, the middle panels show the soil moisture at field capacity, and the right panel shows the estimated percolation on 6th March 2023

Another crucial factor in determining the net water requirement is precipitation. Both Penman-Monteith and SEBAL evapotranspiration models do not inherently account for the effect of precipitation. To estimate the net water requirement accurately, it is essential to incorporate the precipitation events (if any). For this purpose, the precipitation for the current week or the operational week is estimated using the GPM IMERG data. Given the operational objective of the framework, only IMERG's early run data is utilized. The cumulative precipitation for the seven days preceding the latest Landsat overpass date is considered. Additionally, to provide accurate advisories to water managers, the cumulative sum of forecasted precipitation for the next seven days from the Landsat overpass date is also included in the calculations. Consequently, the net water requirement can be determined using the following equation:

Net Water Requirement = 
$$SEBAL\ ET - PET - Per + P_{nc} + P_{fc}$$

Here in Equation 7, SEBAL ET is the estimated evapotranspiration, PET is the evapotranspiration estimated using the Penman-Monteith (proxy of actual water demand of crop), Per is the percolation,  $P_{nc}$  is the nowcast precipitation, and  $P_{fc}$  is the forecasted precipitation.

- 385 Based on the constituents of the above equation, the net water requirement is assessed as follows:
  - If the net water requirement is positive, it indicates that the combined effect of last week's water supply, the amount of percolation that happened (Equation 6), the current week's precipitation, and the next week's forecasted precipitation has met or exceeded the water demand. In this case, the net water requirement for the given command area is considered to be null, as no additional water is needed in the coming week.
  - If the net water requirement is negative, it signifies that the combined effect of the last week's water supply, the amount of percolation happened, the current week's precipitation, and the next week's forecasted precipitation was insufficient to meet the upcoming demand. Consequently, additional water is required for the given command area in the coming week.

395

390

375



It must be remembered that to calculate the surplus or deficit regions with precipitation factored in, we do not require to know the system losses, such as canal seepage, canal evaporation or sluice gate efficiency. This is because our calculations are based on the field conditions after the water has been delivered or consumed on the field. The actual ET estimated from the SEBAL algorithm is assumed to be a proxy of on-field water use or on-field irrigation.

## 5.4. Command Area Irrigation Status (Step 4-5):

After estimating the net water requirements for each command area, sDRIPS evaluates whether the current water supply conditions can meet these demands or not. To perform this evaluation, the water supply at the head canal is proportioned according to the area of each command area. This proportioning is based on the assumption that a larger command area will have more cropped land, resulting in higher evapotranspiration and, consequently, a greater water demand compared to smaller command areas. A distribution factor (DF) is created using the following equation:

$$DF_i = \frac{A_i}{\sum_{i=1}^N A_i}$$

Where DF<sub>i</sub> is the distribution factor for i<sup>th</sup> command area, N is the total number of command areas, A<sub>i</sub> is an area of i<sup>th</sup> command area. After estimating the distribution factor for each command area, the available water supply at the head canal is multiplied by this factor to determine the amount of water distributed to each command area. Subsequently, the amount of water passing through each canal is estimated based on the distribution to the command areas.

Fig. 6 illustrates this distribution process, showing how water is allocated to various command areas within the irrigation canal network. In Fig. 6, let the total water supply at the head canal be denoted as T. The main canal (C) branches into secondary canals (S1C, S2C, and S3C) and further into tertiary canals (T1S2C, T1S3C, T2S3C). The distribution factor (DF) for a command area b is calculated as shown in 9:

$$DF_b = \frac{b}{a+b+c+d+e}$$

The water supplied to command area *b* is then determined by:

$$WS_b = T \times (DF_b)$$

Fig. 6: Illustration of how water distribution( dispatch) advisory for the command areas of an irrigation canal network.

Similarly, the water passing through canal *S2C* includes the amount of water supplied to its command area and the water distributed to its sub-canals (Fig 6). This can be expressed as below:

$$W_{S2C} = WS_b + WS_c$$

# 6. Results



#### 6.1. Water Needs and Distribution in Command Areas

Fig. 7 illustrates the four panels generated by sDRIPS after running it for 14th March 2023 for TBP. For a better understanding of the readers, these panels are discussed in detail as follows. Fig. 7 Panel (a) depicts the main Teesta canal bifurcating into three primary canals and the percentage distribution of water in each canal. Using the DF, it is estimated that the Dinajpur canal receives approximately 27%, the Bogra canal receives around 17%, and the Rangpur canal receives about 18% of the total water supply withdrawn from the Teesta River for that week. Similarly, the water distribution for each canal is also quantified. Overall, the Teesta main Canal, carrying 100% of the water, supplies around 62% to these three primary canals, with the remaining 38% distributed to smaller canals serving command areas upstream of the primary canals. Quantifying this information is beneficial for any canal system (not limited to TBP) with some control over the distribution of water to its secondary and tertiary canals. This allows water managers to monitor and adjust the water distribution to each command area based on current needs. The percentage of

water flowing in each canal was estimated using the basic principles of network theory, where the amount of water from the lowest canal with no bifurcations was summed up to the primary canal.

Fig. 7: (a) Canal water distribution based on the distribution factor where head canal is assumed to have 100% water at the source; (b) amount of water allotted to each command area from the water supply based on (a); (c) net water requirement of each command area after the inclusion of previous week's water supplied to crops, nowcast, and forecast precipitation, and amount of water percolated (d) surplus or deficit command areas after the integrating all the components of water supply (panels b and c).

Fig. 7 Panel (b) illustrates an example of the amount of water that provided to each command area by the water supply from the Teesta River using the canal network. This is calculated by multiplying the DF with the available water supply at the head canal and following the hierarchical canal network (primary to secondary; secondary to tertiary and so on).

https://doi.org/10.5194/egusphere-2025-4574 Preprint. Discussion started: 16 October 2025 © Author(s) 2025. CC BY 4.0 License.

470

490

Fig. 7 panel (c) illustrates the net water requirement, calculated by integrating multiple factors. These factors include the water applied to plants during the previous week, the current water needs of the plants, 455 the precipitation received, the forecasted precipitation for the upcoming week and the amount of total water percolated. This estimation follows 7. The combined effect of these factors, excluding the canal water supply, met the needs of 60% of the command areas for the given date. This information is crucial for the hydrologic community, including water managers and researchers, as it identifies regions dependent on precipitation variability. Over an extended period, water managers and researchers can study the areas that 460 are primarily reliant on precipitation and are sensitive to changes in precipitation patterns. When precipitation is adequate, these regions lessen the stress on water managers. Conversely, insufficient precipitation exacerbates the stress on water resources. Additionally, these regions can vary based on crop type, growth stage of crop, and precipitation intensity. Researchers can leverage this data to study the longterm sensitivity of various crops to precipitation variability, informing crop selection in regions with high 465 precipitation variability and recommending robust crops capable of withstanding such conditions.

Fig. 7 panel (d) is derived by integrating the net water requirements of the command areas (Panel c) with the water provided by the available water supply (Panel b). This panel highlights regions within the TBP that have either surplus or deficit water. This information is crucial for water managers to make informed decisions about adjusting and optimizing water supply across different regions in the upcoming dispatch cycle. To provide a clear understanding to readers, we have discussed the various colored regions depicted in Panel (d) below.

Red and Blue Regions: These areas have either surplus water (blue) or deficit water (red). These regions pose a significant challenge to water managers. To address this, water managers can divert water from surplus areas to nearby deficit areas if the supply infrastructure allows it and there is buy-in from farmers. Additionally, they can also advise farmers in the deficit regions to cultivate less water-intensive crops to balance water distribution more effectively.

480 Light-Colored Regions: These areas exhibit low intensity of surplus or deficit water. Water managers should aim to stabilize the red and blue regions to fall within this range. These regions typically include areas where - i) The water needs were already met by precipitation, and additional water was supplied. ii) The water needs were only partially met by precipitation, but the combined effect of the water supply sufficiently met the requirements, albeit with a slight surplus.

## 485 6.2. Reliability of sDRIPS

To promote the applicability of sDRIPS in other regions where the proposed framework might be suitable and to instill confidence in its reliability among readers, we conducted a two-pronged investigation: i) comparing the estimates of sDRIPS with in-situ data, and ii) comparing the estimates of sDRIPS with OpenET. In the first approach, we evaluated the net water requirement for the entire TBP by aggregating the net water requirement estimates for each command area. These estimates were calculated based on the combined response of the previous irrigation cycle, nowcast and forecast precipitation, and the amount of water percolated, as outlined in Equation 7. For consistency in the analysis, we assumed rice, a water-intensive crop, was cultivated uniformly across all regions of the TBP. We then compared the sDRIPS-derived net water requirement with the actual amount of water supplied through the main canal system of

495 the TBP during the period from January 2023 to April 2023. Additionally, an uncertainty analysis was performed on the net water requirement by introducing perturbations in the parameters of the evapotranspiration models and running Monte Carlo simulations. For more details, readers are referred to Section S3 of the supplementary materials.

### Net Water Required Vs Water Supplied

Fig. 8: Timeseries of net water required by TBP vs amount of water supplied to TBP.

From Fig. 8, we can infer the following:

- Early Crop Stage: At the early stage of crop growth, the water supplied through the TBP canal sufficiently met the crop's water requirements. However, during this period, there can be instances where the water supplied exceeded the crop water needs, indicating potential inefficiencies in water allocation.
- 2. Mid-Crop Stage: During the mid-stage, the water supply from the TBP canal was insufficient to meet the crop water requirements, resulting in unmet water demand.
- 3. Precipitation Effects: Precipitation intensity significantly reduced the water burden on the TBP canal during certain weeks:
  - a. 14th March 2023, 7th April 2023, and 23rd April 2023: Precipitation alone satisfied the crop water needs for the TBP region, demonstrating that the TBP region could have significantly benefitted from integrating precipitation and its forecasting into its water allocation planning. This could reduce unnecessary reliance on canal water during times of sufficient rainfall.
  - b. 30th March 2023: Despite the precipitation, the combined effect of rainfall and TBP water supply was inadequate to meet crop water demands due to low precipitation intensity.
- 4. Late Crop Stage: Toward the end of the crop cycle, both water demand and TBP water supply decreased. During this period, precipitation also occurred, further reducing the canal's burden.

515

500

505

525

530

In the second approach, we compared the sDRIPS estimates (SEBAL ET as implemented within the sDRIPS framework) with the OpenET. OpenET leverages 30m Landsat imagery and employs six state-of-the-art satellite based energy balance models, that is, ALEXI/DisALEXI, eeMETRIC, geeSEBAL, PT-JPL, SIMS and SSEBop (details on these models can be found in Volk et al., (2024)). These models have been extensively applied and evaluated in the United States for various water management and agricultural applications (Volk et al., 2024). According to Volk et al., (2024), a comparison of OpenET data with 152 in-situ datasets across the USA showed that the daily ensemble results for cropland sites had a mean absolute error of 23.6% and an RMSE of 31.1% of the mean. However, OpenET is currently limited to use within the USA. To enable a fair comparison, we applied the sDRIPS framework to regions within the USA where OpenET data is available. Specifically, we selected two locations for comparison, as illustrated in Fig. 9 and Fig. 10

#### Actual Evapotranspiration: OpenET Vs sD.R.I.P.S for California

Fig. 9: Location of the region of interest in California with the timeseries of estimates from OpenET and sDRIPS-based SEBAL ET. Satellite basemap from ESRI.

#### Actual Evapotranspiration: OpenET Vs sD.R.I.P.S for Nebraska

Fig. 10: Location of the region of interest in Nebraska with the timeseries of estimates from OpenET and sDRIPS. Satellite basemap from ESRI.

In both Fig. 9 and Fig. 10, the upper panel shows the location of the region of interest where sDRIPS was applied, and the lower panel shows the timeseries of the estimates. The sDRIPS estimates align closely with the OpenET ensemble range, although some outliers are observed. Excluding the outliers, two key inferences can be drawn from the comparison:

- 1. Both sDRIPS and OpenET exhibit similar trends in ET estimates across the studied regions.
- 2. During peak periods (e.g., July), sDRIPS estimates slightly underpredict ET, or OpenET slightly overpredicts it (noting that OpenET itself is not 100% accurate). This discrepancy is likely due to differences in the spatial resolution of meteorological input data that comes from different sources. OpenET utilizes Gridded Surface Meteorological (gridMET) (Abatzoglou, 2013) and North American Land Data Assimilation System (NLDAS) products (Xia et al., 2012). Both gridMET and NLDAS are limited to the USA with spatial resolution of 4 km and around 13 km respectively;

545







whereas sDRIPS relies on a global (GFS) product with a coarser spatial resolution of 25 km to obtain the meteorological variables (see Table 1).

## 6.3. Comparative Assessment of sDRIPS Scenarios

In this section, we perform a comparative assessment of sDRIPS with the help of two scenarios. In scenario one, we simulate traditional irrigation practices, whereas in scenario two, we simulate scenario one using sDRIPS. More details are provided in the following subsections.

## 6.3.1. First Scenario - Traditional Irrigation Practices (without sDRIPS)

In this scenario, we illustrate traditional irrigation practices from the perspectives of both water providers and water consumers. The traditional practice from the water provider's perspective involves allotting water without knowledge of the specific needs of command areas and without accounting for precipitation. From the water consumers' perspective, traditional irrigation involves watering crops based solely on the water allocated to them, again without considering precipitation. It is important to remember that the TBP operates mostly from January to April, prior to the next monsoon season. During this period, farmers do not account for precipitation events in their irrigation practices. We illustrate the scenario simulation using the panels shown in Fig. 11.

Fig. 11: Scenario 1 example without the s.D.R.I.P.S - Simulation of traditional irrigation practices i.e. without understanding and quantifying the actual water need of the crop and command area.

Panel (a) of Fig. 11 shows the water requirements estimated for each command area for rice crops using the FAO-recommended Penman-Monteith equation. Panel (b) of Fig. 11 depicts the amount of water applied to each command area by farmers (for the week of March 14, 2023), including water added from precipitation events. However, farmers do not anticipate these precipitation events and fully utilize the allotted water, disregarding previous irrigation cycle and precipitation. Panel (c) of Fig. 11 illustrates the excess water present in the fields, calculated as the difference between the total water applied to the command area and the Penman-Monteith based water requirements of the crops. The positive values in Panel (c) of Fig. 11 indicate that excess water has been used, which is unsustainable as it can wash away vital nutrients needed by crops, ultimately affecting yield.






## 6.3.2. Second Scenario - Adaptive Irrigation Practice (With sDRIPS)

In this scenario, we consider the same time frame as in Scenario 1 and for the same week; however, here, water providers use the sDRIPS system. This system accounts for water applied in the previous irrigation cycle (and whether it was sufficient or deficit), current precipitation events, and forecasted precipitation. Water is allocated based on the actual needs of the command areas, considering these factors. The simulation is illustrated with the panels shown in Fig. 12.

Fig. 12: Illustration of Scenario 2 with the s.D.R.I.P.S - Simulation of adapting irrigation practices i.e. understanding and quantifying the actual water need of the crop and command area.

Panel (a) of Fig. 12 is similar to Panel (a) of Fig. 11 and is estimated using the Penman-Monteith equation. Panel (b) of Fig. 12 shows the combined response, considering the previous irrigation cycle, the recent precipitation event, and forecasted precipitation before the next irrigation cycle. Including these factors reduces the stress on water providers and guides them to allocate water only where it is needed. Panel (c) of Fig. 12 illustrates the amount of water applied by the farmers. The water was allocated to each command area specific to its needs and was fully utilized by the farmers.

By comparing Scenario 1 (traditional irrigation practices) and Scenario 2 (optimized irrigation with sDRIPS), approximately 475 acre-feet of water could have been saved using sDRIPS for the specific week as an example. The saved water could then be diverted to water deficit regions outside the TBP or stored for future TBP needs. This comparison focused on March 14, 2023, when the available water supply exceeded the water demand of the region. However, there are instances where the net demand exceeds the total water supply, as illustrated in the panels of Fig. 13 for February 25, 2023.

Panel (a) of Fig. 13 shows the Penman-Monteith based water needs for each command area. Panel (b) of Fig. 13 depicts the combined response, including the effects of the previous irrigation cycle and precipitation. Panel (c) of Fig. 13 illustrates the water applied to the command areas with the limited water supply. Panel (d) of Fig. 13 presents the water stress map, highlighting regions where water demands were not met. In situations where the water demand exceeds the supply, if water providers or canal operators have greater control over the irrigation canals, they should reconsider the DF of the command areas. This would allow for the redistribution of water from surplus regions to deficit regions. If such control is not feasible, water providers should either manage the water needs by diverting water from outside the TBP to required command areas or encourage farmers to grow less water intensive crops in those regions.

Fig. 13: sDRIPS simulation for 25th March 2023, when the net water demand was higher than the water supply.

# 7. User Friendliness of sDRIPS and Limitations

## 7.1. User Friendliness

The sDRIPS system is designed to address the gaps identified in the current literature, as mentioned in section 1 of the study, by providing a user-friendly and customizable tool for analyzing various irrigation scenarios. These are discussed in detail in Section S2 of the supplementary material.








#### 7.2. Limitations

- While sDRIPS provides a scalable and data-driven framework for surface water irrigation optimization, it is not without limitations. These limitations arise from sensor constraints, practical assumptions required for scalability, and certain challenges that lie outside the scope of this study. The following key limitations are acknowledged:
  - sDRIPS relies on optical sensors Landsat series, to estimate actual ET using SEBAL. High cloud cover, common in certain regions or seasons, can compromise the framework's effectiveness by hindering the energy balance.
    - 2. sDRIPS identifies crop fields using the Copernicus Global Land Cover dataset, though land cover maps may not always fully capture dynamic changes over time, potentially affecting net water requirement estimates. However, the framework is designed to be flexible, allowing users to refine or integrate higher-resolution local land cover data, when available, to better distinguish between irrigated and rainfed areas for improved water management.
    - 3. Groundwater contributed minimally or negligibly to crop evapotranspiration during the study period due to the considerable depth of the water table relative to crop root lengths. However, in regions with shallower water tables or highly conductive soils, groundwater may influence evapotranspiration estimates. Current state-of-the-art satellite-based techniques often lack the spatial resolution needed to precisely quantify groundwater contributions at the farm scale. Integrating model-based simulations or in situ groundwater flow measurements into the framework's water balance equations could enhance estimation accuracy and provide a more comprehensive understanding of water dynamics.
    - 4. The framework assumes stable climatic conditions over a weekly timescale to estimate the cumulative ET. While this assumption aligns with regional climatology in the study area and is supported by other studies (Bose et al., 2021), it may not hold true in regions where climatology exhibits high variability within a week. Consequently, users should carefully evaluate the applicability of this assumption when deploying sDRIPS in areas with high dynamic climatic conditions.
    - 5. The framework does not account for the behavior of water-providers and water-consumers, which can significantly influence the effectiveness of any irrigation planning scheme. Behavioral change, particularly among farmers, is a gradual and long-term process that falls outside the direct scope of this study. While this study demonstrates the utility of a satellite data-driven surface water irrigation optimization tool, stakeholder-led interventions, such as economic incentives (e.g., price differentiation), may be necessary to accelerate behavior change, as discussed by Portoghese et al., (2021). Furthermore, behavioral dynamics can vary across different regions and countries, necessitating tailored approaches for each context.
    - 6. Hydraulic constraints are assumed to be invariant in the study. Users can integrate the sDRIPS framework with different hydraulic designs and can check the flexibility (see Sawassi et al., 2022).
    - 7. The assumption that water percolation below the 100 mm depth is lost to deep percolation may lead to an overestimation of irrigation needs, especially for crops that are known to utilize water beyond this depth. This simplification is a conservative approach that prioritizes caution in estimating crop water requirements, which is particularly important in water-scarce regions where slight overestimating water needs could lead to more effective water management decisions. However, it

also introduces uncertainty, and future work could refine this aspect by incorporating models that predict root zone soil moisture or by using satellite-based soil moisture estimates for deeper layers. Additionally, methodologies such as those proposed by (Baldwin et al., 2017) could be explored to improve the estimation of water availability in deeper soil layers.

## 8. Conclusion

In this study, we co-developed and implemented a water-provider-centric irrigation advisory system, sDRIPS, to manage water resources sustainably and robustly while addressing existing gaps in the literature. sDRIPS integrates satellite and model data to estimate the actual water needs of command areas. This crucial information aids water providers in determining the appropriate water allocation for each command area. The actual water need depends on several factors, including the amount of water applied in the previous irrigation cycle, current precipitation, and forecasted precipitation. By considering these factors and the available water supply, sDRIPS advises canal operators on which command areas require water and the amount needed. During periods of excess water supply, sDRIPS advises canal operators on how much water can be stored for future irrigation needs or diverted to water-deficit regions outside the irrigation project. Conversely, during water shortages, sDRIPS generates water stress maps to identify regions experiencing water shortages and guides on the amount of water needed in deficit areas. sDRIPS also allows for the simulation of various scenarios using historical data on different crops. This capability enables stakeholders to evaluate the impact of future policies on water supply conditions if these policies are implemented today and to devise science-based responses. By offering a flexible and comprehensive tool for water management, sDRIPS has the potential to contribute significantly to sustainable irrigation practices globally, addressing the challenges posed by a changing climate and inefficiency in managing surface water and groundwater resources.

685

690




#### Data and software availability:

The sDRIPS framework can be accessed at <a href="https://github.com/UW-SASWE/sDRIPS">https://github.com/UW-SASWE/sDRIPS</a>. The in-situ dataset provided by the BWDB is not publicly uploaded due to its sensitive nature but can be made available upon request.

#### **Author contributions:**

SK: research design, investigation, software, and writing; FH: research design, investigation, and writing; KI: data resources, MA: data resources



#### **Competing interests:**

The contact author has declared that none of the authors has any competing interests.

**Acknowledgements:** The study was supported by NASA Applied Science (Earth Science to Action) Water Resources grant 80NSSC22K0918.

725

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
