# Peer review of "Satellite Data Rendered Irrigation using Penman-Monteith and SEBAL - sDRIPS for Surface Water Irrigation Optimization"

_EGUsphere, 2025_

## Author Comment (AC1)

**Answer to Reviewers' Comments**

We thank the editor for the overall guidance. We also thank the reviewers for taking the time to review our manuscript and offer constructive comments and wisdom for improvement. We have considered all the comments and addressed the questions raised by the reviewers to improve our work presented in the paper. In the following text, the reviewer's questions/comments are listed in black font and the response by the authors in blue font. All line number references mentioned in this response to reviewers refer to the revised version of our manuscript.

As part of the revision process, we have substantially revised the manuscript. Detailed, point-by-point responses to each reviewer are provided below, with a brief summary of the major revisions included at the beginning of each reviewer's section.

**Reviewer #1**

The authors propose a framework for providing spatially distributed irrigation advices by leveraging Earth Observation, which is for sure a timely topic in the broader context of water resources management. However, three main issues critically affect the manuscript quality: We sincerely thank Reviewer #1 for the thorough and constructive review of our manuscript, as well as acknowledging the timeliness of the study. Overall, reviewers' comments primarily focused on:

1) the clarification of the study period,
2) validation or providing any referenced study in the region,
3) manuscript length
4) clarification on the estimation of soil moisture.

To address the reviewer's point, we have substantially revised the manuscript by:

1. Expanding and clarifying the soil moisture estimation, including a correction on the scaling approach used to convert relative soil moisture to volumetric estimates.
2. Providing the scientific validation of the study, along with the referenced studies using the Surface Energy Balance Algorithm (SEBAL) in similar agroclimatic settings, including Bangladesh.
3. Shortening the introduction section of the manuscript.

Below, we have provided detailed answers to each comment raised by the reviewer.

1. Implementation over just few days/weeks. This is a gap, at least an entire irrigation season

should be covered to prove the method robustness. Fig. 8 is where a 3-month period is shown for the first time, but it is not clear whether it is a whole season or not. Since water allocation managers often have control on seasonal volumes, I believe that comparing seasonal-aggregated advices with availabilities is meaningful.

We thank the reviewer for this constructive comment. We agree that demonstrating robustness over a full operating season is important, and we clarify here that our study already covers the entire irrigation period of the Teesta Barrage Project (TBP).

The TBP provides canal-based surface water irrigation only during the dry months, i.e, January through April. Outside this period, fields in the command area rely on rainfall, and TBP (whose purpose is for dry-season irrigation) does not operate. Accordingly, the period shown in Fig. 8 represents the complete operational irrigation season for TBP rather than an exploratory interval.

Although the TBP operating season is described in the first paragraph of Section 2 (Study Area), we appreciate the reviewer's concern regarding clarity for readers. To make this explicit, we have revised the caption of Fig. 8 as follows:

*"Fig. 8. Time series of net water required by TBP vs. water supplied. TBP operates from January to April each year; outside this period, fields rely mainly on rainfall."*

2.    Validation: even though I appreciate authors' efforts in this sense, it is not carried out in a rigorous way. It targets other areas (with different climate conditions) and likely different vegetation/crop patterns. In addition, SEBAL, which is the basis for ET calculation for sDRIPS, is involved in OpenET also. If a direct validation is not possible, authors should just look for and mention previous studies validating SEBAL in several climate context (possibly involving the one of interest) to support the method reliability.

We thank the reviewer for this constructive comment. While direct field-scale validation within the TBP command area was not feasible due to the absence of lysimeter or flux-tower measurements, we have strengthened the manuscript by explicitly referencing validation studies of SEBAL and Penman-Monteith ET in agroclimatic conditions comparable to our study region.

Notably, the study by Bose et al. (2021), which underpins the original development of the sDRIPS framework, conducted lysimeter-based validation at the Bangladesh Agricultural Research Institute (BARI) in Gazipur from 2017 to 2018. Their results demonstrated:

- strong correlation between Penman-Monteith estimates and lysimeter observations ($r \approx$ 0.87-0.88; errors 2-10% for wheat/maize), and
- SEBAL errors ranging from ~8% to 46%, which Bose et al. (2021) largely attributed to pixel-point mismatch and mixed land cover near the lysimeter site.

[Figure]

*Fig. 1: Taken from supplementary material Figure S1 of Bose et al. (2021).*

[Figure]

*Fig. 2: Taken from supplementary material Figure S2 of Bose et al. (2021).*

Importantly, Bose et al. (2021) relied on Landsat 7 thermal data (100 m), whereas the current study uses Landsat 8/9 thermal bands (30 m), which substantially reduce mixed-pixel effects and are expected to improve ET estimation accuracy.

Consistent with Bose et al. (2021), several independent studies in comparable agroclimatic settings have also demonstrated strong agreement between SEBAL-derived ET and lysimeter observations, as shown in Table 1 below.

Table 1: Studies validating SEBAL ET with lysimeters in a similar agroclimatic setting.

| Sr. No | Region | Sensor Used | Metrics for comparison between SEBAL ET and Lysimeter ET | Source |
|--------|--------|-------------|-----------------------------------------------------------|--------|
| 1. | Pakistan | Landsat 8 TIR | RMSE = 1.26 mmd$^{-1}$, $R^2$ = 0.9, Nash-Sutcliffe Efficiency (NSE = 0.92) | Zahid et al., 2023 |
| 2. | Bangladesh | Landsat 8 TIR | $R^2$ = 0.99, RMSE = 0.16 mmd$^{-1}$ | Islam et al., 2023 |
| 3. | India | Landsat 7 TIR | $R^2$ = 0.91 | Rawat et al., 2017 |

Altogether, these studies in similar agroclimatic regions and also in Bangladesh provide strong empirical support for the reliability of SEBAL as well as Penman-Montieth ET. In the manuscript, we have added the aforementioned references on the validation of SEBAL and Penman-Monteith for similar agroclimatic regions.

3. Manuscript organization: the paper is too long and a bit confused in several parts.

Although we have shortened the Introduction section in the revised manuscript, we note that a substantial portion of the manuscript length is attributable to essential visual material, including a table summarizing the datasets used (approximately one page) and thirteen figures that together occupy a significant fraction of the manuscript. These elements are included to clearly document the data sources, methodology, and results that are directly relevant to the study.

Please            find            additional            comments            as            follows:

L 90-94: some initiatives to fill this gap are under development, e.g., ESA (European Space Agency) WorldCereal (https://esa-worldcereal.org/en)

We thank the reviewer for highlighting this relevant initiative. To acknowledge ongoing efforts addressing this gap and to further inform readers, we have incorporated a reference to the ESA WorldCereal program (Franch et al., 2022) in the Introduction section of the revised manuscript (Line 97 of the updated manuscript).

L 138-147: it concerns data used, should not be in the introduction

We agree with the reviewer's comment. The description of the datasets previously included in Lines 138-147 has been removed from the Introduction, as these details are already summarized in Table 1 of the manuscript.

L 149-186: should be sharply shortened

We thank the reviewer for this suggestion. The section corresponding to Lines 149-186 has been shortened in the revised manuscript (now Lines 140-175) to improve conciseness and readability.

L 198-204: keep units' consistency and use the International System
We have now updated the lines with the SI units. The lines now read as the following:

*"The sDRIPS study focuses on the Teesta Barrage Project (TBP) in Bangladesh, the country's largest surface water irrigation project. Established in 1990, the TBP supplies irrigation water from January to April and spans the Teesta River at Dalia-Doani Point in the Lalmonirhat district (Fig. 1 Middle Panel). This project features a 615-meter-long concrete structure equipped with 44 radial gates, providing a discharge capacity of 361 $m^3$/s (12,750 cfs) and supporting a command area of $1.5425x10^9$ $m^2$ (154,250 hectares) through a 4,500-kilometer network of canals (River Research Institute, 2023). For more details on TBP, readers are referred to (River Research Institute, 2023)."*

Eq 2: here is an underlying assumption that evaporation from bare soil is negligible, the Kc you are calculating is basically Kcb. If so, please detail on this. A question on top of this. At this stage soil moisture is already needed to compute Ks, isn't it?
We agree that under the formulation used in Eq. (2), the crop coefficient corresponds to the basal crop coefficient (Kcb) rather than the full crop coefficient (Kc). In the context of the TBP, irrigation operations occur predominantly during the dry season (January-April), when rainfall events are infrequent, and soil evaporation following wetting is minimal. Under these conditions, Kc ≈ Kcb, consistent with the FAO-56 framework.

Furthermore, we would like to reiterate that the reliability of this formulation is supported by the in situ lysimeter-based evaluation conducted by Bose et al. (2021), which used the above coefficients and demonstrated the accuracy of both Penman-Monteith (where crop coefficients are utilized) and SEBAL evapotranspiration under similar agroclimatic conditions.

Estimation of soil moisture using Sentinel 1 is already covered in the manuscript in a later section.

L 334-336: sounds like results

We appreciate reviewers' comments and have relocated this line to the results section (Line 495-499 of the updated manuscript).

Fig 4: it seems that some coarse resolution input drives this result

We acknowledge the comment and note that, in any modeling framework (not limited to sDRIPS), the use of coarse resolution inputs can influence outputs generated at finer spatial scales; sDRIPS is not an exception in this regard. The sDRIPS framework relies on a subset of meteorological forcing variables obtained from the Global Forecast System (GFS), as summarized in Table 1 of the updated manuscript. These globally available and near-real-time datasets are provided at a coarser spatial resolution, reflecting a common trade-off between spatial resolution, global coverage, and low latency in operational meteorological products.

If higher spatial resolution, globally consistent meteorological datasets with comparable latency were available publicly on the cloud, they could be readily incorporated into the sDRIPS framework. Therefore, this limitation arises from the current characteristics of global meteorological datasets rather than from the sDRIPS methodology itself. This limitation and its implications have now been explicitly acknowledged and discussed in the revised manuscript (Lines 675-679).

*"sDRIPS relies on meteorological forcings from the GFS, which has relatively coarse spatial resolution. This limitation may affect ET and water requirement estimates in regions with strong spatial heterogeneity. In the TBP region, however, irrigated fields are extensive and relatively homogeneous, reducing the sensitivity of results to coarse resolution meteorological inputs."*

Soil Moisture: how was it calculated? This is an important omission. In addition, it is expressed as water height (mm). This implies that the reference volume is known. Soil layer is fixed at 100 mm (as seen by Figures), what about the porosity?

We thank the reviewer for bringing this to our attention. In the manuscript, we stated that soil moisture was estimated from Sentinel 1 C-band SAR and used to quantify percolation (Eq. 5–6), but we did not sufficiently describe (i) the Sentinel 1 soil moisture retrieval framework employed and (ii) how soil moisture was expressed in millimeters.

We clarify here that soil moisture is first retrieved at the field scale from Sentinel 1 IW GRD SAR backscatter using the established change-detection framework developed by Bauer-Marschallinger et al., (2019), which is a widely used and validated approach for Sentinel 1 surface soil moisture retrieval and underpins several large scale and operational soil moisture products (Bauer-Marschallinger et al., 2019; Copernicus, 2018; Rahmati et al., 2025).

Importantly, no novel soil moisture retrieval methodology was developed in this study; rather, we implemented this existing framework, and readers are referred to Bauer-Marschallinger et al.

(2019) and Meyer et al. (2022) for full methodological details. In brief, the Sentinel 1 backscatter value $\sigma^0$ (t) at time t is normalized to dry and wet reference states observed in the past, yielding relative surface soil water saturation SSM(t) in percent as follows:

$$SSM(t) = \frac{\sigma^0(t) - \sigma^0{}_{dry}}{\sigma^0{}_{wet} - \sigma^0{}_{dry}}$$

Because this change detection approach yields relative soil moisture, we subsequently scale the Sentinel 1 derived signal to volumetric soil moisture (m³/m³) using Soil Moisture Active Passive (SMAP) volumetric soil moisture observations as a reference, following the methodology described in Bauer-Marschallinger et al., (2019) and Meyer et al., (2022). We note that, in the earlier version of the manuscript, the scaling step required to convert relative soil moisture to volumetric soil moisture was not explicitly implemented; this has now been. After implementing this correction, we found no change in the overall conclusions of the study, although minor quantitative differences required updates to several figures.

Volumetric soil moisture is converted to an equivalent water depth (mm) by multiplying by an effective soil depth Z. Given that Sentinel 1 C-band observations (wavelength ≈ 5 cm) are primarily sensitive to the near-surface soil layer and have demonstrated reliable performance up to approximately 100 mm depth, we set Z = 100 mm:

$$SM_{mm} = f_{SMAP}(SSM(t)) \times Z$$

Regarding soil physical properties, porosity is not explicitly required in this conversion because volumetric soil moisture already represents the fraction of pore space occupied by water. However, soil hydraulic properties are required to determine soil water storage limits. Accordingly, field capacity was obtained from the ISRIC SoilGrids dataset (https://isric.org/explore/soilgrids, Poggio et al., 2021), using the volumetric water content at -33 kPa matric potential, which is widely accepted as a proxy for field capacity. SoilGrids-derived hydraulic properties have been extensively used in large-scale hydrological and agricultural modeling studies and are readily available within Google Earth Engine. Grid level estimates were aggregated to the command area scale using the median and multiplied by irrigated area to estimate net water requirements.

To avoid over-lengthening the manuscript and diverting attention from the main methodological contribution, we have added a concise description of the Sentinel 1 soil moisture estimation, scaling to SMAP, and conversion to water depth in the revised Data and Methods section, while explicitly referring readers to Bauer-Marschallinger et al. (2019) and Meyer et al. (2022) for full technical details.

Eq 6: maybe you could directly mention this one instead of eq 5 – which is a specific case of eq 6
We appreciate the constructive feedback, and we have now directly mentioned it in Eq. 6 (Eq. 6 of the previous manuscript, now Eq. 7 of the submitted manuscript), and have removed the separate presentation of Eq. 5 (Eq. 5 of the previous manuscript) to improve clarity and avoid redundancy.

Fig 5: How did you handle the soil moisture estimate at the command area level?
We have addressed this in the "How soil moisture was estimated" comment.

Eq 7: even though it is explained afterwards, this equation could be misleading, as rainfall reduces net water requirement
We respectfully disagree with this point, as Eq. 7 clearly mentions the rainfall component, and also accounts for it, which is explained in the subsequent paragraph.

L 399-400: this should be proved

We thank the reviewer for pointing this out. However, this line was also mentioned in the introduction with the reference (now Line 145-146 of the revised manuscript).

L 474-484: not needed

We thank the reviewer for this suggestion. After careful consideration, we have chosen to retain these lines as the content is relevant to the interpretation of the results.

L 495: more details on this in the "Data" Section?

We thank the reviewer for this suggestion. We have now added the in situ data to the table in the Data section and updated it to include other datasets that were previously missed.

Fig 9 and 10: On top of the criticism to the significance of validation, description here is qualitative only, as no metrics are provided

We thank the reviewer for their constructive feedback. In the revised manuscript, we have complemented the qualitative discussion of Figs. 9 and 10 with quantitative performance metrics (Lines 540–548). For the California region (Fig. 9), sDRIPS shows strong temporal agreement with the OpenET ensemble mean (r = 0.72), with an RMSE of 1.37 mm/d, MAE of 0.99 mm/d, and a moderate negative bias of -0.68 mm/d. The corresponding Kling-Gupta efficiency (KGE = 0.59) indicates good overall agreement when jointly accounting for correlation, bias, and variability.

[Figure]

*Fig. 3: OpenET Timeseries along with sDRIPS estimates for the studied region in California*

For the Nebraska region (Fig. 10), agreement is stronger, with a correlation coefficient of r = 0.86, RMSE of 1.25 mm/d, MAE of 0.88 mm/d, and a bias of -0.48 mm/d. The KGE value of 0.50 likewise reflects good consistency between the two products. We emphasize that these comparisons are intended as benchmarking and consistency assessments, rather than independent validation, as the input datasets in the OpenET are relatively finer than input datasets in sDRIPS.

[Figure]

*Fig. 4: OpenET Timeseries along with sDRIPS estimates for the studied region in Nebraska*

**References for Reviewer 1:**

Bauer-Marschallinger, B., Freeman, V., Cao, S., Paulik, C., Schaufler, S., Stachl, T., Modanesi, S., Massari, C., Ciabatta, L., Brocca, L., & Wagner, W. (2019). Toward Global Soil Moisture Monitoring With Sentinel-1: Harnessing Assets and Overcoming Obstacles. *IEEE Transactions on Geoscience and Remote Sensing*, *57*(1), 520–539. IEEE Transactions on Geoscience and Remote Sensing. https://doi.org/10.1109/TGRS.2018.2858004

Copernicus. (2018). *Surface Soil Moisture 2014-present (raster 1 km), Europe, daily – version 1* [Dataset]. https://doi.org/10.2909/e934b15f-7d48-4c6d-a9c6-6484488aa58f

Franch, B., Cintas, J., Becker-Reshef, I., Sanchez-Torres, M. J., Roger, J., Skakun, S., Sobrino, J. A., Van Tricht, K., Degerickx, J., Gilliams, S., Koetz, B., Szantoi, Z., & Whitcraft, A. (2022). Global crop calendars of maize and wheat in the framework of the WorldCereal project. *GIScience & Remote Sensing*, *59*(1), 885–913. https://doi.org/10.1080/15481603.2022.2079273

Islam, A. T., Islam, A. S., Islam, G. T., Bala, S. K., Salehin, M., Choudhury, A. K., Dey, N. C., & Mahboob, M. G. (2023). Simulation of water productivity of wheat in northwestern Bangladesh using multi-satellite data. *Agricultural Water Management*, *281*, 108242. https://doi.org/10.1016/j.agwat.2023.108242

Meyer, R., Zhang, W., Kragh, S. J., Andreasen, M., Jensen, K. H., Fensholt, R., Stisen, S., & Looms, M. C. (2022). Exploring the combined use of SMAP and Sentinel-1 data for downscaling soil moisture beyond the 1 km scale. *Hydrology and Earth System Sciences*, *26*(13), 3337–3357. https://doi.org/10.5194/hess-26-3337-2022

Poggio, L., de Sousa, L. M., Batjes, N. H., Heuvelink, G. B. M., Kempen, B., Ribeiro, E., & Rossiter, D. (2021). SoilGrids 2.0: Producing soil information for the globe with quantified spatial uncertainty. *SOIL*, *7*(1), 217–240. https://doi.org/10.5194/soil-7-217-2021

Rahmati, M., Balenzano, A., Bechtold, M., Brocca, L., Fluhrer, A., Jagdhuber, T., Karamvasis, K., Mengen, D., Reichle, R. H., Kim, S., Taghizadeh-Mehrjardi, R., Walker, J., Zhu, L., & Montzka,

C. (2025). Soil moisture retrieval from Sentinel-1: Lessons learned after more than a decade in orbit. *Remote Sensing of Environment*, *333*, 115146. https://doi.org/10.1016/j.rse.2025.115146

Rawat, K. S., Bala, A., Singh, S. K., & Pal, R. K. (2017). Quantification of wheat crop evapotranspiration and mapping: A case study from Bhiwani District of Haryana, India. *Agricultural Water Management*, *187*, 200–209. https://doi.org/10.1016/j.agwat.2017.03.015

Zahid, M. N., Ahmad, S., Khan, J. A., Arshad, M. D., Azmat, M., & Ukasha, M. (2023). Evapotranspiration estimation using a satellite-based surface energy balance: A case study of Upper Bari Doab, Pakistan. *Environmental Earth Sciences*, *82*(24), 601. https://doi.org/10.1007/s12665-023-11284-5

---

## Author Comment (AC2)

**Answer to Reviewers' Comments**

We thank the editor for the overall guidance. We also thank the reviewers for taking the time to review our manuscript and offer constructive comments and wisdom for improvement. We have considered all the comments and addressed the questions raised by the reviewers to improve our work presented in the paper. In the following text, the reviewer's questions/comments are listed in black font and the response by the authors in blue font. All line number references mentioned in this response to reviewers refer to the revised version of our manuscript.

As part of the revision process, we have substantially revised the manuscript. Detailed, point-by-point responses to each reviewer are provided below, with a brief summary of the major revisions included at the beginning of each reviewer's section.

**Reviewer #2**

The authors have developed a satellite remote sensing-based method of estimating irrigation requirements at the field application scale – utilizing the Penman-Monteith and the SEBAL methods of evapotranspiration – and the sDRIPS approach to provide irrigation advisory for surface water resources allocation.

The manuscript is very well-written, with a thorough literature review, visualizations, and rationale of the applications approach. The mathematical equations and details of the step-by-step approach for irrigation requirements calculation is appreciated. The results with figures and the detailed discussion on the limitations are noted as well.

We sincerely thank Reviewer #2 for the thorough and constructive review of our manuscript, as well as for the positive assessment of the clarity, methodological rigor, visualizations, and discussion of limitations in the manuscript. Overall, reviewers' comments were on

1) clarification on figures,
2) the source of the data,
3) the most appropriate units for representing irrigation information

We have addressed these points in the revised manuscript, and detailed, point-by-point responses to each comment are provided below.

1. Lines 84-86: For the 'How much' of irrigation – isn't the unit depth of irrigation (L) a better estimate for irrigation requirements (also widely used metric across the world)? From the water depth required, the water volume ($L^3$) and the flow rate ($L^3/T$) can easily be calculated based on field size and hydraulic infrastructure?

We partially agree with the reviewer's comment. We concur that irrigation depth is a widely used and intuitive metric for expressing irrigation requirements, particularly at the field and farm-management scales. The proposed framework fundamentally estimates irrigation requirements in terms of an equivalent water depth, consistent with established agronomic practice. This depth based estimate is subsequently converted to volumetric units ($L^3$) based on the irrigated area.

The emphasis on volumetric units in the manuscript is intentional and reflects the intended operational context of the framework. Specifically, volumetric representations are more suitable for surface water allocation and system-level decision-making, such as at the canal-command scale, where water delivery is typically managed in terms of total volume to be supplied. Furthermore, the in-situ water supply data used for comparison were provided by stakeholders in volumetric units rather than depth.

The choice of metric is therefore user-dependent. For farmers, field-level advisories expressed as irrigation depth are directly actionable. In contrast, canal operators and water managers are responsible for allocating water across multiple fields within a canal command area (including downstream and secondary canals) and therefore require estimates of aggregate water volume rather than depth. Importantly, depth, volume, and flow-rate metrics are readily interconvertible based on field area and locally available hydraulic characteristics.

2. Lines 360, 366: In Equations 5 and 6, how is 'Field Capacity' estimated? It is unclear.

Field capacity is not directly estimated within the sDRIPS framework. Instead, sDRIPS relies on externally derived soil hydraulic properties obtained from global soil datasets.

In the revised manuscript, we clarify that field capacity is obtained from the ISRIC SoilGrids dataset (https://isric.org/explore/soilgrids, Poggio et al., 2021) as the volumetric soil water content at a matric potential of ~ -33 kPa, which is widely accepted as a proxy for field capacity (Hengl et al., 2017; Poggio et al., 2021).

To improve clarity, we have updated Table 1 in the Data section (of the revised manuscript) to explicitly document this data source.

3. The visualizations of Figure 2 are excellent. The gridded 'Surplus/Balanced/Deficit Regions' approach are not, however, found later. Is Figure 4 a modified version of this approach? Are the gridded calculations aggregated at the field scale in later figures?

We thank the reviewer for the positive feedback on Figure 2 and for the insightful question regarding the relationship between the conceptual figure and the subsequent figures.

Figure 2 is intended to present the conceptual and computational foundation of sDRIPS for any general region, illustrating how evapotranspiration-based water balance components are first computed at the pixel (grid) scale. At this stage, each pixel is classified as surplus, balanced, or deficit based on evapotranspiration and precipitation water balance.

Figure 4 does not represent a separate or modified approach. Rather, it provides a spatially explicit example of the same grid-based framework illustrated in Figure 2, applied to a specific command area within the Teesta Barrage Project (TBP) for a particular date.

In subsequent figures, the reviewer is correct that these grid-level calculations are aggregated to higher spatial scales. Specifically, pixel-level surplus and deficit estimates are first aggregated to the field scale and then at the command-area scale to support operational water allocation decisions. Overall, Figure 2 presents the general grid-based conceptual framework, Figure 4 demonstrates its application for a specific region and date, and the subsequent figures illustrate the aggregation of this grid-based information to management-relevant scales.

**References for Reviewer 2:**

Hengl, T., Jesus, J. M. de, Heuvelink, G. B. M., Gonzalez, M. R., Kilibarda, M., Blagotić, A., Shangguan, W., Wright, M. N., Geng, X., Bauer-Marschallinger, B., Guevara, M. A., Vargas, R., MacMillan, R. A., Batjes, N. H., Leenaars, J. G. B., Ribeiro, E., Wheeler, I., Mantel, S., & Kempen, B. (2017). SoilGrids250m: Global gridded soil information based on machine learning. *PLOS ONE*, *12*(2), e0169748. https://doi.org/10.1371/journal.pone.0169748

Poggio, L., de Sousa, L. M., Batjes, N. H., Heuvelink, G. B. M., Kempen, B., Ribeiro, E., & Rossiter, D. (2021). SoilGrids 2.0: Producing soil information for the globe with quantified spatial uncertainty. *SOIL*, *7*(1), 217–240. https://doi.org/10.5194/soil-7-217-2021